# Determinants of Stunting Among Children Aged 0.5 to 12 Years in Peninsular Malaysia: Findings from the SEANUTS II Study

**DOI:** 10.3390/nu17142348

**Published:** 2025-07-17

**Authors:** Ika Aida Aprilini Makbul, Giin Shang Yeo, Razinah Sharif, See Meng Lim, Ahmed Mediani, Jan Geurts, Bee Koon Poh

**Affiliations:** 1Faculty of Health Sciences, Universiti Kebangsaan Malaysia, Kuala Lumpur 50300, Malaysia; ikaaprlini@gmail.com (I.A.A.M.); jasonyeo1993@hotmail.com (G.S.Y.); smlim@ukm.edu.my (S.M.L.); 2Institute of Systems Biology, Universiti Kebangsaan Malaysia, Bangi 43600, Malaysia; ahmed@ukm.edu.my; 3FrieslandCampina, 3818 LE Amersfoort, The Netherlands; jan.geurts@frieslandcampina.com

**Keywords:** child malnutrition, ethnicity, maternal health, Malaysia, stunting, Southeast Asia

## Abstract

**Background/Objectives**: Childhood stunting remains a critical public health issue in low- and middle-income countries. Despite Malaysia’s economic growth, there is limited large-scale evidence on the determinants of stunting among children from infancy to primary school age. This cross-sectional study, part of South East Asian Nutrition Surveys II (SEANUTS II), aimed to determine sociodemographic and environmental risk factors for stunting among 2989 children aged 0.5–12 years. **Methods**: Children were recruited from four regions in Peninsular Malaysia (Central, East Coast, 2022–2030Northern, Southern). Standing height or recumbent length was measured, and stunting was classified based on WHO criteria (height-for-age Z-score below −2 standard deviations). Parents reported information on socioeconomic status, sanitation facilities, and hygiene practices. Multivariate binary logistic regression was used to determine the determinants of stunting. **Results**: Stunting prevalence was 8.9%, with infants (aOR = 2.92, 95%CI:1.14–7.52) and young children (aOR = 2.92, 95%CI:1.80–4.76) having higher odds than school-aged children. Key biological predictors included low birth weight (aOR = 2.41; 95%CI:1.40–4.13) and maternal height <150 cm (aOR = 2.24; 95%CI:1.36–3.70). Chinese (aOR = 0.56; 95%CI:0.35–0.88) and Indian children (aOR = 0.16; 95%CI:0.05–0.52) had a lower risk of stunting compared to Malays. **Conclusions**: This study highlights the ongoing challenge of childhood stunting in Malaysia, with age, birth weight, ethnicity, and maternal height identified as key determinants. These findings call for early identification of at-risk households and targeted support, especially through education and financial aid to foster healthy child growth.

## 1. Introduction

Stunting, a condition characterized by impaired linear growth due to prolonged malnutrition, remains a pressing public health challenge in Malaysia. Defined as a height-for-age Z-score below −2 standard deviations (SD) of the World Health Organization (WHO) growth standards [1], stunting is associated with various long-term consequences, including diminished cognitive development, increased vulnerability to chronic diseases, and reduced economic productivity in adulthood [2,3]. Stunting remains a significant global issue, affecting around 22% of children worldwide in 2020, with an estimated 149.2 million cases [4]. In Malaysia, 21.8% of children under the age of five years are stunted, with rural and indigenous communities disproportionately affected [5].

Tackling stunting is critical not only for improving individual health, but also for breaking the intergenerational cycle of poverty and achieving national development goals in alignment with the National Plan of Action for Nutrition of Malaysia (NPANM) III [6]. Despite its status as an upper-middle-income country, the phenomenon of stunting in Malaysia remains a cause for concern and reflects the complex factors that contribute to its persistence. While economic progress has reduced poverty, disadvantaged groups continue to be affected by gaps in access to healthcare, maternal education, and dietary practices [7]. Additionally, variations in living conditions, cultural practices, and geographic location further contribute to disparities in nutritional status among children, particularly in under-resourced communities [8,9,10]. Understanding how socioeconomic and maternal determinants interact to influence stunting rates is critical for developing strategies that align with Malaysia’s commitment to equitable health outcomes under the Sustainable Development Goals (SDGs) [11].

Despite extensive research on stunting in children in Malaysia, there are still significant gaps in knowledge. Most studies focus on children under the age of five [12,13,14,15], while older children, who may still have growth deficits, are hardly considered [16,17]. While parental education and household income are recognized as influential factors, their relative importance remains unclear, calling for further research [16,18]. The complexity of the determinants of stunting is also compounded by differences in access to healthcare, maternal employment, and economic conditions between urban and rural areas [15,19,20]. For instance, working mothers in urban areas often rely on processed foods for their own meals due to time constraints, which can affect the quality of children’s diets [21].

The existing literature indicates that several modifiable risk factors are associated with an increased risk of stunting, including inadequate dietary intake [18,22], suboptimal breastfeeding and complementary feeding practices [23,24], and maternal characteristics such as poor nutritional status [25,26], low education level [23,27], and parental smoking [23,28]. Contextual factors, such as poor sanitation, inadequate hygiene, and limited access to healthcare services [17,29], have also been linked to impaired child growth. In addition, chronic infections and inflammation, which are often associated with poor living conditions and repeated exposure to enteric pathogens, can compromise nutrient absorption and further hinder growth [30,31]. These risk factors not only influence early childhood development but may also have lasting effects beyond the age of five. However, existing studies have often examined these factors independently or within limited age ranges, particularly focusing on children under five [12,13,14,15], resulting in an incomplete understanding of how these determinants collectively influence growth across the full span of childhood.

Addressing these gaps is critical to developing comprehensive life-cycle approaches that support children’s growth from infancy through adolescence. This study examines the sociodemographic and environmental factors influencing stunting among children aged 0.5 to 12 years in Malaysia, filling an important research gap by including older children who have often been excluded in previous studies. By assessing child and parental characteristics, such as maternal height, employment status, education level, household income, household size, as well as household and environmental factors, including parental smoking habits and sanitation and hygiene practices, this study provides an integrated analysis of how these determinants affect child growth across different socioeconomic and environmental contexts. The results will serve as a basis for tailored strategies, such as school feeding initiatives and caregiver training programs, to promote long-term growth and development.

## 2. Materials and Methods

### 2.1. Study Design

This study used data from the South East Asian Nutrition Surveys II (SEANUTS II), which included 2989 healthy Malaysian children aged 6 months to 12 years. The study was conducted from May 2019 to March 2020 in the four main regions of Peninsular Malaysia: Central, Southern, Northern, and East Coast regions. Participants were selected using a multistage cluster sampling procedure to ensure a representative sample from the regions studied. Sampling was conducted in two stages. First, the Department of Statistics Malaysia (DOSM) randomly selected two districts, one urban and one rural, from different states, within each region. In the second stage, a list of randomly selected enumeration blocks within these districts was provided by DOSM for participant recruitment. Children living in households or educational institutions located within a 5 km radius of the selected enumeration blocks were eligible for inclusion. Children aged 0.5–6 years were recruited through home visits, while those aged 7–12 years were recruited through a school-based approach. For younger children attending preschools, data collection was also conducted at their nurseries or kindergartens. The detailed study design and protocols have been previously published [32,33]. For this analysis, data included a total of 2973 children, after excluding those with implausible or invalid data on relevant variables (n = 16).

### 2.2. Ethical Approval and Permission for Data Collection

SEANUTS II has been formally registered in the Dutch Trial Registry (NL7975). All procedures were conducted in accordance with the principles outlined in the Declaration of Helsinki and received ethical approval from the Universiti Kebangsaan Malaysia Research Ethics Committee (JEP-2018-569). Permission for data collection was obtained from the Ministry of Education Malaysia, state education departments, and other relevant agencies, including the Department of Community Development (KEMAS), school principals, kindergartens, nurseries, and community leaders. Prior to data collection, written informed consent was obtained from parents or guardians, and verbal assent was obtained from participating children.

### 2.3. Sociodemographic Status and Children’s Health Status and Environmental Factors Questionnaire

Two structured questionnaires were used for data collection, which were available in bilingual form (Malay–English or Mandarin–English) and were completed by the parents or guardians. These were the questionnaire on sociodemographic status and the questionnaire on children’s health status and environmental factors. The sociodemographic status questionnaire collected data on key demographic variables such as sex, age, ethnicity, area of residence, and birth weight of the child. Information was also collected on parental characteristics such as age, height, weight, level of education, occupation, and total household income. The questionnaire on environmental factors included those that are known to affect children’s health, such as source of drinking and cooking water, sanitation, waste disposal methods, hand hygiene, and parental smoking status. The questions on environmental factors were adapted from the WHO’s Core Questions on Drinking Water and Sanitation for Household Surveys [34] and previous research [35].

### 2.4. Anthropometric Measurement

All anthropometric measurements were carried out by trained researchers according to standardized procedures. Recumbent length of children under two years of age was measured to the nearest 0.1 cm using either the SECA 210 measuring mat or SECA 417 Infantometer (Seca GmbH, Hamburg, Germany). For older children, standing height was recorded to the nearest 0.1 cm using a SECA 213 stadiometer (Seca GmbH, Hamburg, Germany). To ensure measurement accuracy, the maximum allowable difference between repeated measurements was set at less than 0.5 cm for both recumbent length and standing height. Anthropometric status was classified using the WHO growth standards 2006 for children aged 0 to 5 years [36] and the WHO growth reference 2007 for children aged 5 to 19 years [37]. These standards are internationally recognized for age-appropriate assessment of child and adolescent growth. These standards support standardization in growth assessment and enable consistent comparisons across regional and national populations. Z-scores for height-for-age were calculated using raw anthropometric measurements. For children under 5 years, WHO Anthro version 3.1.0 was used [38], while for children aged 5 years and older, WHO AnthroPlus version 1.0.3 was applied [39]. Children were classified as “stunting” if the height-for-age Z-score (HAZ) value was < −2 SD, while children with a Z-score above −2 SD were classified as normal height.

### 2.5. Statistical Analysis

Statistical analyses were performed with IBM SPSS Statistics for Windows version 22.0 (IBM Corp., Armonk, NY, USA), using the complex sampling module. To ensure nationally representative estimates for children aged 0.5–12 years, a sample weighting factor was derived by taking into account the national distribution by region, area, sex, ethnicity, and age, based on the 2019 projected population following the 2010 Malaysian Census [40]. Descriptive statistics (weighted proportions, means, standard errors, and 95% confidence intervals [CI]) characterized sociodemographic and anthropometric variables. Independent variables with *p* < 0.05 in the univariate analyses (Appendix A) were included in the final multivariable model, while those with *p*-values between 0.05 and 0.10 were included for adjustment purposes. Multivariate binary logistic regression identified the determinants of stunting using two models: (1) unadjusted odds ratios (OR) and (2) adjusted odds ratios (aOR) controlling for residential area (urban/rural) and parental smoking status. Results are presented as OR/aOR with 95%CI, and statistical significance was assessed with two-tailed tests (*p* < 0.05).

## 3. Results

### 3.1. Child and Maternal Characteristics Related to Stunting

The study identified several demographic and socioeconomic factors associated with stunting among children aged 0.5–12 years in Peninsular Malaysia (n = 2973), as detailed in Table 1. The overall prevalence of stunting among children aged 0.5–12 years was 8.9% (n = 231), with a mean height-for-age Z-score (HAZ) of −2.45 ± 0.03. The prevalence of stunting differed significantly between age groups (*p* < 0.001), with higher rates observed in younger children (0.5–3.9 years) compared to older cohorts. Biological factors, particularly low birth weight (<2.5 kg), were significantly more prevalent among children with stunting than in children without stunting (*p* < 0.001). Ethnicity was also found to be significantly associated with stunting, with Malay children exhibiting a higher prevalence compared to children of Chinese and Indian ethnicities (*p* < 0.01). Additionally, children with stunting were more likely to have siblings and come from larger households with five or more family members (*p* < 0.05). In contrast, no significant differences were observed between stunting and sex or urban–rural residence. Parental factors, particularly maternal age and height, were significantly associated with child stunting. Children of mothers aged ≥40 years had a low rate of stunting compared to those with mothers aged 30–39 years (*p* < 0.01). Additionally, stunting was more prevalent among children from households with short-statured mothers (*p* < 0.01).

### 3.2. Sanitation and Hygiene Practices of the Children and Mothers

Sanitation and hygiene practices were largely similar between children with and without stunting in Peninsular Malaysia (Table 2). Nearly all households had access to improved drinking water and cooking water, regardless of children’s stunting status, with no cases of stunting reported in the few households that did not have these resources. A high proportion of both stunted and non-stunted groups had access to improved sanitation facilities, with children in both groups having similar levels of access. Children’s hygiene practices, including handwashing before meals or after toilet use, soap use, and wearing shoes outdoors, showed no notable differences between the two groups. Likewise, maternal handwashing before food preparation and after toilet use was similar among mothers of children with and without stunting. Given that there were no significant differences between groups, these environmental factors were not included in the final regression models. However, parental smoking status was retained as an adjustment variable due to its potential confounding effect.

### 3.3. Determinants of Stunting Among Children

Multivariate analysis identified significant determinants of stunting among children aged 0.5–12 years in Peninsular Malaysia (Table 3). Age was significantly associated with stunting risk: infants (0.5–0.9 years) had 2.92 times the odds of being stunted (95%CI: 1.14–7.52; *p* < 0.05), while those aged 1.0–3.9 years were 2.92 times at risk of stunting (95%CI: 1.80–4.76; *p* < 0.001), compared to school-aged children (7.0–12.9 years). Biological factors further contributed to stunting: low birth weight (below 2.5 kg) was associated with twice the odds (aOR = 2.41; 95%CI: 1.40–4.13; *p* < 0.01), and maternal height less than 150 cm similarly increased the risk of stunting by 2.24 times (95%CI: 1.36–3.70; *p* < 0.01). There were significant ethnic differences, with Chinese (aOR = 0.56; 95%CI: 0.35–0.88; *p* < 0.05) and Indian children (aOR = 0.16; 95%CI: 0.05–0.52; *p* < 0.01) being less likely to be stunted as compared to Malay children. Household structure also influenced outcomes: children residing in larger households (5 members or more) were 1.54 times at risk of stunting (95%CI: 1.03–2.31; *p* < 0.05) in the unadjusted model only. Similarly, having 1 to 2 siblings increased the risk of stunting by 1.63 times compared to those with no siblings (95%CI: 1.05–2.53; *p* < 0.05) in the unadjusted model only.

## 4. Discussion

This study identifies the key determinants of stunting among children aged 0.5–12 years in Peninsular Malaysia, contextualizing findings within the broader landscape of Malaysian child health research. The 8.9% stunting prevalence observed in this study is lower than the 21.8% reported in the National Health and Morbidity Survey (NHMS) for children under the age of five [5]. Similarly, SEANUTS II reported a slightly lower prevalence of 13.8% for the same age group compared to NHMS [32]. This discrepancy in stunting prevalence observed in younger children may be influenced by differences in study design, sampling methods, and population characteristics, as NHMS covered the entirety of Malaysia, while SEANUTS II was conducted in Peninsular Malaysia, as the data collection in Bornean Malaysia has been halted due to COVID-19 outbreaks. However, the overall prevalence among school-aged children in our study is consistent with NHMS data, which also shows lower stunting rates in this age group (12.7% among those aged 5–17 years) [5]. Additionally, the first SEANUTS Malaysia study recorded a stunting prevalence of 8.4% among children aged 0.5–12 years [41], which is comparable to SEANUTS II. This indicates minimal progress over time despite ongoing public health initiatives.

In addition to stunting, this study acknowledges the importance of considering other forms and indicators of undernutrition. Although not the primary focus of the present analysis, additional indicators such as weight-for-age Z-score (WAZ) and weight-for-height Z-score (WHZ) provide complementary insights into the broader nutritional status of children. According to SEANUTS II, among children under five years, the mean WAZ was –0.72 ± 0.05 and mean WHZ was –0.38 ± 0.05, with 11.4% of children classified as underweight and 6.2% wasted, suggesting that younger children may be at greater nutritional risk [32]. These findings, previously reported in the main SEANUTS II findings, underscore the complexity of child undernutrition and highlight the relevance of integrated, multi-sectoral approaches to address the range of nutritional challenges among children in Malaysia.

The persistently higher stunting rates among infants and toddlers (0.5–3.9 years) aligns with NHMS findings, reinforcing early childhood as a critical period for intervention even in middle-income settings. In this study, younger children, particularly those under the age of 4 years, were significantly more likely to be stunted compared to older children, highlighting early childhood as a critical window of vulnerability to growth faltering. Stunting is usually less common among older children, suggesting that growth may improve over time [42,43]. Another possible explanation is the increased access to nutrition and healthcare support through formal education among older children. In Malaysia, for instance, all children are required to complete six years of primary education under mandatory education policies as outlined in the Malaysia Education Act 1996. Underprivileged children enrolled in government schools are more likely to be identified and directly supported through various school-based government food assistance programs, including school meal programs, milk supplementation initiatives, and food basket distributions [44]. In contrast, younger children who are not yet attending school rely primarily on household engagement with healthcare services to access support. Although the food assistance program, such as food baskets, is available for this age group and is intended to benefit the entire household, its direct impact on the child may be uncertain, as usage cannot be monitored at home and is only assessed during follow-up visits. These findings underscore early childhood, particularly the period before school age, as a critical stage of vulnerability to stunting, highlighting the need for improved monitoring and more targeted interventions to ensure that younger children receive adequate and effective nutritional support.

In addition, children with low birth weight (<2.5 kg) and those born to mothers with a height below 150 cm were more likely to be stunted, consistent with previous findings from both Malaysian and global studies [45,46,47]. Shorter maternal stature has been linked to an increased risk of stunting in children, as it often reflects a history of undernutrition that can contribute to fetal growth restrictions and hinder postnatal development [45,48]. Studies have also found that this association is particularly evident in South Asia, where maternal height is frequently an indicator of early-life nutritional challenges in children, even in countries undergoing economic transition [49,50]. Additionally, research suggests that shorter women are more likely to have narrower pelvic structures, which can impact the uterine environment, potentially restricting fetal growth and increasing the likelihood of low birth weight [51]. Our findings further align with the existing literature linking shorter maternal height to childhood stunting, underscoring the influence of maternal anthropometry on early growth outcomes [52,53]. This association highlights the importance of maternal nutrition and health interventions in improving child growth and reducing the risk of undernutrition.

Similarly, low birth weight is a well-recognized risk factor for stunting, as it can result from suboptimal intrauterine conditions that may affect early growth and development [54]. Previous studies have also shown that low birth weight infants are more prone to growth faltering, largely due to restricted fetal development and insufficient nutritional stores at birth [55,56]. This association is concerning because low birth weight infants are at higher risk of undernutrition during crucial growth phases, which can make it challenging for them to catch up in growth. Additionally, the link between maternal height and low birth weight further supports the intergenerational cycle of undernutrition, as shorter mothers are more likely to have smaller infants who remain at risk of stunting throughout childhood [47,55].

Ethnic differences in stunting prevalence persisted in this study, with the risk of stunting higher in Malay children than their Chinese or Indian peers. This finding aligns with NHMS 2019 data showing elevated stunting rates in rural, Malay-majority regions [5]. Ethnicity-related dietary behaviors may partly explain this disparity. Among Malaysian ethnic groups, Malays have been reported to adopt a mix of Westernized and local dietary patterns, characterized by frequent breakfast skipping and high intakes of rice, fried foods, and sweetened beverages, which may contribute to suboptimal nutrition [57]. In contrast, Chinese individuals are more likely to follow dietary habits aligned with health-promoting guidelines, such as higher consumption of vegetables, fruits, and lean proteins [57]. In Malaysia’s multiethnic setting, cultural practices strongly influence food choices and meal habits, which can in turn affect nutritional outcomes.

Although this study did not directly assess dietary intake, findings on dietary patterns from SEANUTS II Malaysia have been published separately by Hasmuni Chew et al. (2024) [8]. The results suggest that traditional dietary patterns prevalent in Malay communities, such as reliance on rice-based diets with limited diversity in protein and micronutrient sources, may contribute to these disparities [8,58,59]. Dietary intake plays a crucial role in childhood growth, with suboptimal nutrient consumption contributing to stunting risk [60]. In Malaysia, inadequate protein intake has been linked to higher stunting prevalence, particularly among children in rural areas [13]. Studies have also shown that low dietary diversity and insufficient consumption of nutrient-dense foods, such as animal-source proteins, fruits, and vegetables, exacerbate growth deficits, reinforcing the need for targeted nutritional interventions for Malaysian children [13,60]. However, the absence of dietary data analysis in the present study limits the ability to explore these associations in depth, highlighting the need for future research to quantify dietary diversity and nutrient intake across ethnic groups, with particular attention to complementary feeding practices, especially among children under two years old.

Household structure, including larger households (≥5 members) and the presence of siblings, was linked to increased likelihood of stunting, which could be a sign of resource dilution, a phenomenon in which the available nutritious food is overly distributed among family members, leading to inadequate nutrition. While different studies have used varying household size thresholds to define “large” households, the association between large household size and stunting has been widely observed in low-resource settings [61,62], with related patterns observed in the Malaysian context [18]. Having siblings may increase competition for limited resources, which can influence stunting rates by reducing individual access to essential nutrients and care in larger households [63]. While our analysis did not specifically examine differences between the oldest and youngest siblings, we acknowledge that factors, such as family resource allocation and short birth intervals, may influence nutritional status, highlighting a need for further research in this area.

Unlike previous studies that have linked poverty to stunting, this study found no significant associations between household income and maternal employment and stunting rates. This contrasts with other findings where socioeconomic factors, including income and maternal employment status, have been shown to influence child nutrition and growth [16,64]. While socioeconomic indicators, such as household income and maternal employment, showed limited direct association with stunting in this study, the existing literature underscores the potential role of maternal education and healthcare access in shaping child nutrition [21,65]. Although maternal education did not emerge as a significant factor in this study, it is generally recognized as a crucial determinant of child health outcomes, with educated mothers more likely to implement healthy dietary practices and access better healthcare services [66]. Similarly, no association was observed between sanitation and hygiene practices and stunting, which may reflect generally adequate access within the study population.

The strengths of this study include a nationally representative sample of children aged 0.5 to 12 years in Peninsular Malaysia, which was analyzed using robust survey-weighted methods to account for the complex sampling design. By integrating biological (e.g., maternal height, birth weight), demographic (e.g., ethnicity, household structure), and socioeconomic variables, it provides a multifactorial perspective on stunting rarely explored in the context of upper-middle-income countries. However, the cross-sectional design of the study precludes causal conclusions, and the fact that the data are based on self-reporting leads to a possible recall bias. Key confounders, such as dietary diversity and healthcare access, were not assessed, potentially overlooking important factors influencing stunting. Despite these limitations, the findings highlight the shifting determinants of stunting in transitioning economies and underscore the need for longitudinal, equity-focused research to address residual disparities in underserved communities. This is in line with global priorities, such as SDG 3 (Good Health and Well-being) and SDG 10 (Reduced Inequalities) by addressing disparities in vulnerable populations [11].

While the study contributes important insights, several limitations should be acknowledged when interpreting the factors linked to stunting. It should consider other relevant factors in addition to anthropometry, such as dietary intake, breastfeeding practices, micronutrient status, and markers of infection or inflammation. These factors are known to influence child growth but were not included in this analysis due to data limitations. Future research should address these limitations by incorporating longitudinal designs and comprehensive measures of nutritional, environmental, and biological determinants to advance our understanding of stunting etiology.

## 5. Conclusions

The findings indicate that early childhood (<4 years) represents a high-risk period for stunting, with infants and toddlers being 2.9 times more likely to be stunted compared to school-aged children, highlighting the continued importance of interventions in the first 1000 days of life. Maternal height <150 cm and low birth weight were found to be key biological predictors, highlighting the intergenerational aspects of malnutrition and the potential value of prenatal programs for at-risk women. Ethnic differences, with Malay children at higher risk than their Chinese or Indian peers, highlight cultural and systemic inequities, suggesting a need for localized, community-driven nutrition strategies. These observations suggest that in Malaysia, biological and cultural factors need to be considered alongside broader socioeconomic measures in measures to combat stunting. Malaysia has adopted a multi-sectoral approach to address malnutrition through a variety of national initiatives outlined in the NPANM III and targeted strategies under the National Strategic Plan to Address Stunting (2022–2030), which emphasize early childhood nutrition, maternal health, and equity-based approaches. These include educational campaigns, school-based interventions, and the implementation of regulatory guidelines aimed at promoting healthy eating habits. Complementing these initiatives, the Malaysian Dietary Guidelines for Children and Adolescents were recently updated to offer specific, evidence-based guidance on healthy eating behaviors, targeting both knowledge and practical application among the younger population. To sustain progress, future longitudinal research should evaluate these interventions while exploring causal pathways between maternal health, dietary diversity, and intergenerational stunting. Collaborative efforts involving policymakers, healthcare providers, and local communities will be critical to achieving Malaysia’s SDG targets and ensuring equitable health outcomes.

## Figures and Tables

**Table 1 nutrients-17-02348-t001:** Child and maternal characteristics related to stunting.

Characteristics	All (n = 2973)	Non-Stunted (n = 2742)		Stunted (n = 231)		Chi-Square Value	*p* Value
	Unweighted Count (%)	Unweighted Count (%)	95%CI	Unweighted Count (%)	95%CI		
**All, n (%)**	2973	2742 (91.1)	89.7–92.3	231 (8.9)	7.7–10.3		
Recumbent length/standing height (cm), mean ± SE	115.17 ± 0.57	116.76 ± 0.59		98.93 ± 1.82			
Height-for-age Z-scores (HAZ) (Mean ± SE)	−0.58 ± 0.02	−0.40 ± 0.02		−2.45 ± 0.03			
** *Child Characteristics* **							
**Age group (Years)**							
Infants (0.5–0.9 years)	76 (3.9)	65 (84.0)	70.9–91.9	11 (16.0)	8.1–29.1	52.172	<0.001 ***
Toddlers (1.0–3.9 years)	555 (23.9)	479 (85.6)	81.9–88.7	76 (14.4)	11.3–18.1		
Preschoolers (4.0–6.9 years)	919 (24.5)	850 (91.1)	88.3–93.3	69 (8.9)	6.7–11.7		
School-aged children (7.0–12.9 years)	1423 (47.7)	1348 (94.4)	92.6–95.7	75 (5.6)	4.3–7.4		
**Sex**							
Girls	1555 (48.5)	1433 (90.7)	88.6–92.4	122 (9.3)	7.6–11.4	0.598	0.552
Boys	1418 (51.5)	1309 (91.5)	89.4–93.2	109 (8.5)	6.8–10.6		
**Residential areas**							
Rural	881 (26.6)	793 (89.2)	86.4–91.5	88 (10.8)	8.5–13.6	4.565	0.085
Urban	2092 (73.4)	1949 (91.8)	90.1–93.2	143 (8.2)	6.8–9.9		
**Ethnicity**							
Malay	1782 (74.4)	1599 (89.5)	87.7–91.1	183 (10.5)	8.9–12.3	28.009	0.008 **
Chinese	867 (15.8)	827 (95.1)	93.2–96.5	40 (4.9)	3.5–6.8		
Indian	279 (6.2)	273 (97.9)	94.5–99.2	6 (2.1)	0.8–5.5		
Others	45 (3.6)	43 (94.6)	76.7–98.9	2 (5.4)	1.1–23.3		
**Birth Weight (n = 2660)**							
Low (<2.5 kg)	251 (10.2)	213 (82.6)	75.4–88.1	38 (17.4)	11.9–24.6	24.974	<0.001 ***
Normal (2.5 kg and above)	2409 (89.8)	2237 (91.8)	90.3–93.1	172 (8.2)	6.9–9.7		
**Household size (n = 2964)**							
<5 people	908 (30.8)	845 (93.2)	91.0–94.9	63 (6.8)	5.1–9.0	7.343	0.025 *
5 or more people	2056 (69.2)	1888 (90.1)	88.2–91.7	168 (9.9)	8.3–11.8		
**Number of Siblings (n = 2945)**							
No siblings	1088 (35.2)	1017 (93.7)	91.7–95.2	71 (6.3)	4.8–8.3	13.121	0.017 *
1–2 siblings	1383 (45.8)	1272 (89.7)	87.3–91.7	111 (10.3)	8.3–12.7		
3 or more siblings	474 (19.0)	427 (89.5)	85.6–92.4	47 (10.5)	7.6–14.4		
**Parent Characteristics**							
**Maternal Age Group (n = 2927)**							
<30.0 years	287 (11.1)	265 (91.3)	85.9–94.7	22 (8.7)	5.3–14.1	20.097	0.003 ******
30.0–39.9 years	1715 (57.6)	1560 (89.2)	87.1–91.0	155 (10.8)	9.0–12.9		
>39.9 years	925 (31.3)	873 (94.4)	92.4–95.9	52 (5.6)	4.1–7.6		
**Maternal Height (n = 2876)**							
<150 cm	213 (8.3)	176 (81.3)	74.3–86.7	37 (18.7)	13.3–25.7	29.954	<0.001 ***
≥150 cm	2663 (91.7)	2475 (91.9)	90.4–93.1	188 (8.1)	6.9–9.6		
**Maternal Employment Status (n = 2933)**							
Not Working	1073 (37.5)	980 (90.3)	87.8–92.4	93 (9.7)	7.6–12.2	1.091	0.421
Working	1860 (62.5)	1724 (91.5)	89.6–93.0	136 (8.5)	7.0–10.4		
**Maternal Educational level (n = 2943)**							
Non-schooling/primary school	116 (3.5)	106 (90.1)	81.9–94.9	10 (9.9)	5.1–18.1	0.401	0.868
Secondary school	1391 (47.1)	1280 (91.3)	89.3–93.0	111 (8.7)	7.0–10.7		
Tertiary school (College/University)	1436 (49.4)	1326 (90.7)	88.5–92.6	110 (9.3)	7.4–11.5		
**Total Monthly Household Income** ^a^ **(n = 2916)**							
B40 (≤MYR 4850)	1683 (59.8)	1537 (90.9)	89.0–92.4	146 (9.1)	7.6–11.0	3.052	0.421
M40 (MYR 4851–10,959)	995 (31.9)	927 (90.7)	87.8–93.0	68 (9.3)	7.0–12.2		
T20 (≥MYR 10,959)	238 (8.3)	226 (94.1)	88.8–97.0	12 (5.9)	3.0–11.2		
**Monthly Household Food Expenditure** ^b^ **(n = 2895)**						3.401	0.156
<MYR 783	1217 (47.8)	1107 (90.0)	87.7–91.9	110 (10.0)	8.1–12.3		
≥MYR 783	1678 (52.2)	1563 (92.0)	90.0–93.5	115 (8.0)	6.5–10.0		
**Parental Smoking (n = 2896)**							
No	1677 (54.2)	1558 (92.1)	90.3–93.6	119 (7.9)	6.4–9.7	4.871	0.089
Yes	1219 (45.8)	1112 (89.8)	87.3–91.8	107 (10.2)	8.2–12.7		

^a^ Source: Household Income and Basic Amenities Survey Report 2019, Department of Statistics Malaysia; USD 1 dollar = MYR 4.24 (as of 16 July 2025); B40: low income; M40: middle income; T20: high income. ^b^ Source: Household Expenditure Survey Report 2019, Department of Statistics Malaysia. Significant differences were determined using Complex Samples Pearson Chi-square tests at * *p* < 0.05, ** *p* < 0.01, and *** *p* < 0.001.

**Table 2 nutrients-17-02348-t002:** Sanitation and hygiene practices of the children and mothers.

Parameter	All (n = 2973)	Non-Stunted (n = 2742)		Stunted (n = 231)		Pearson Chi-Square Value	*p* Value
	Unweighted Count	Unweighted Count (%)	95%CI	Unweighted Count (%)	95%CI		
** *Sanitation Factor* **							
**Improved Drinking Water ^a,#^ (n = 2971)**							
No	11 (0.4)	11 (100.0)	100.0–100.0	0	0	-	-
Yes	2960 (99.6)	2729 (91.0)	89.6–92.3	231 (9.0)	7.7–10.4		
**Improved Cooking Water ^b,#^ (n = 2971)**							
No	10 (0.3)	10 (100.0)	100.0–100.0	0	0	-	-
Yes	2961 (99.7)	2730 (91.1)	89.6–92.3	231 (8.9)	7.7–10.4		
**Improved Sanitation Facilities ^c^ (n = 2970)**							
No	311 (9.5)	289 (90.8)	85.2–94.5	22 (9.2)	5.5–14.8	0.025	0.906
Yes	2659 (90.5)	2450 (91.1)	89.6–92.4	209 (8.9)	7.6–10.4		
** *Children’s Hygiene Practices* **							
**Washing hands before meals and after using the toilets (n = 2935)**						0.239	0.668
No	106 (2.7)	98 (89.5)	79.0–95.1	8 (10.5)	4.9–21.0		
Yes	2829 (97.3)	2607 (91.1)	89.6–92.4	222 (8.9)	7.6–10.4		
**Washing hands using soap (n = 2901)**						2.665	0.239
No	380 (11.2)	352 (88.7)	82.9–92.7	28 (11.3)	7.3–17.1		
Yes	2521 (88.8)	2322 (91.4)	89.9–92.7	199 (8.6)	7.3–10.1		
**Wearing shoes outside the house (n = 2950)**						0.374	0.595
No	2564 (88.4)	2368 (91.1)	89.6–92.5	196 (8.9)	7.5–10.4		
Yes	386 (11.6)	352 (90.1)	86.0–93.2	34 (9.9)	6.8–14.0		
** *Mothers’ handwashing practices* **							
**Before preparing food (n = 2969)**						0.135	0.761
No	236 (6.6)	221 (90.4)	84.1–94.3	15 (9.6)	5.7–15.9		
Yes	2733 (93.4)	2517 (91.1)	89.6–92.4	216 (8.9)	7.6–10.4		
**After using the toilets (n = 2967)**						0.124	0.758
No	148 (3.8)	139 (92.0)	84.1–96.1	9 (8.0)	3.9–15.9		
Yes	2819 (96.2)	2597 (91.0)	89.6–92.3	222 (9.0)	7.7–10.4		

^a^ Improved water sources include household connections, public standpipes, boreholes, protected dug wells, protected springs, and rainwater collection (WHO 2023). ^b^ Improved cooking water includes boiling and using filtered or purified water (WHO 2023). ^c^ Improved sanitation includes flush or pour-flush to piped sewer system, septic tank pit latrines, ventilated-improved pit latrines, or pit latrines with slab or composting toilets (WHO 2023). ^#^ Violating Pearson Chi-Square Assumptions.

**Table 3 nutrients-17-02348-t003:** Determinants of stunting.

Parameter	OR	95%CI	*p*-Value	aOR ^#^	95%CI	*p*-Value
**Age group**						
0.5–0.9 years	2.87	1.11–7.44	0.030	2.92	1.14–7.52	0.026 *
1.0–3.9 years	2.89	1.79–4.67	<0.001	2.92	1.80–4.76	<0.001 ***
4.0–6.9 years	1.52	0.92–2.49	0.101	1.54	0.93–2.54	0.092
7.0–12.9 years	1.00			1.00		
**Ethnicity**						
Chinese	0.55	0.35–0.85	0.008	0.56	0.35–0.88	0.012 *
Indian	0.14	0.04–0.46	0.001	0.16	0.05–0.52	0.002 **
Others	0.63	0.09–4.49	0.641	0.64	0.10–4.38	0.651
Malay	1.00			1.00		
**Birth Weight**						
Low (<2.5 kg)	2.51	1.47–4.27	0.001	2.41	1.40–4.13	0.001 **
Normal (2.5 kg and above)	1.00			1.00		
**Maternal Age Group**						
<30.0 years	0.90	0.36–2.24	0.813	0.83	0.32–2.13	0.698
30.0–39.9 years	1.42	0.83–2.42	0.202	1.37	0.79–2.36	0.263
>39.9 years	1.00			1.00		
**Maternal Height**						
<150 cm	2.30	1.39–3.79	0.001	2.24	1.36–3.70	0.002 **
≥ 150 cm	1.00			1.00		
**Household size**						
5 or more people	1.54	1.03–2.31	0.035	1.50	1.00–2.27	0.053
<5 people	1.00			1.00		
**Number of Siblings**						
1–2 siblings	1.63	1.05–2.53	0.030	1.54	0.98–2.42	0.059
3 or more siblings	1.44	0.76–2.72	0.264	1.37	0.71–2.64	0.342
No siblings	1.00			1.00		

^#^ Adjusted for residential area and parental smoking. Multivariate Binary Logistic Regressions: * *p* < 0.05, ** *p* < 0.01 and *** *p* < 0.001.

## Data Availability

Data generated or analyzed during this study are not publicly available due to confidentiality and compliance with European General Data Privacy Regulations. Data are, however, available from the authors upon reasonable request and with permission from the project funder.

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
