# Peer review of "Determinants of Stunting Among Children Aged 0.5 to 12 Years in Peninsular Malaysia: Findings from the SEANUTS II Study"

_nutrients, 2025, doi:10.3390/nu17142348_

Round 1

Reviewer 1 Report

Comments and Suggestions for Authors

This manuscript presents findings from the SEANUTS II survey, offering relevant and updated insights into the determinants of stunting among children in Peninsular Malaysia. The authors apply appropriate statistical tools to a large, representative sample, contributing to both local and regional public health understanding. The topic aligns well with the journal's scope and provides implications for national nutritional strategies. However, several recommendations are suggested:

Major Comments:

  1. Study Design and Sampling

    • The use of a multistage stratified random sampling design is a strength. However, the authors should provide more information on sampling weights, if used, to clarify representativeness.

    • Please explain how households and individual children were selected and whether any form of clustering adjustment was applied in the analyses.

  2. Definition and Classification of Stunting

    • The authors use WHO growth references to define stunting, which is appropriate. However, it's important to specify which WHO growth standards were applied (e.g., 2006 or 2007) and why.

    • Also clarify whether z-scores were calculated using raw anthropometric data or pre-processed by the WHO Anthro software.

  3. Data Analysis

    • The logistic regression is appropriate for binary outcomes, but the selection process for included variables in the final model is not sufficiently described. Were variables chosen based on theoretical relevance or univariate significance?

    • Consider reporting model fit statistics (e.g., Hosmer–Lemeshow test, AUC) to support the robustness of the regression models.

  4. Interpretation of Results

    • While several determinants are statistically significant, the authors should better distinguish between correlation and causation in the discussion.

    • The interpretation of lower odds of stunting in older children needs clarification, especially in light of potential cumulative growth retardation over time.

  5. Policy Implications

    • The conclusion makes brief mention of the relevance to nutrition intervention programs. This could be expanded by referring to existing Malaysian policies or regional initiatives that could benefit from these findings.

Minor Comments

  • In Table 1, please define abbreviations in footnotes (e.g., BMI-for-age z-score).

  • A few typographical errors were found (“underweight” is occasionally misspelled).

  • Consider improving the clarity of some sentences, especially in the results section where percentages and odds ratios are presented together—breaking them into separate statements can improve readability.

Author Response

1. Summary

2. Point-by-point response to Comments and Suggestions for Authors

Overall Reviewer Comment:

This manuscript presents findings from the SEANUTS II survey, offering relevant and updated insights into the determinants of stunting among children in Peninsular Malaysia. The authors apply appropriate statistical tools to a large, representative sample, contributing to both local and regional public health understanding. The topic aligns well with the journal's scope and provides implications for national nutritional strategies. However, several recommendations are suggested:

Comments 1:

Study Design and Sampling

a)      The use of a multistage stratified random sampling design is a strength. However, the authors should provide more information on sampling weights, if used, to clarify representativeness.

b)      Please explain how households and individual children were selected and whether any form of clustering adjustment was applied in the analyses.

Response 1:

Thank you for highlighting this point. Although the sampling was conducted in clusters, no clustering adjustment was applied in the analysis. Instead, a post-stratification adjustment using population weighting factors was applied to ensure national representativeness. This weighting accounted for region, sex, ethnicity, and age distributions, based on the 2019 projected Malaysian population derived from the 2010 Census data.

We have revised the Methods section to provide a clearer description of the sampling stages. Kindly note that the full methodological details have been published elsewhere and have been referenced in the revised sentence to enhance clarity [Page 3, Lines 103 to 112]. Additionally, we have clarified the weighting factor in the Statistical Analysis section. [Page 4, Line 161-164].

Comments 2:

Definition and Classification of Stunting

a)        The authors use WHO growth references to define stunting, which is appropriate. However, it's important to specify which WHO growth standards were applied (e.g., 2006 or 2007) and why.

b)        Also clarify whether z-scores were calculated using raw anthropometric data or pre-processed by the WHO Anthro software.

Response 2:

a)        Revised sentence as suggested for clarity. [Page 4, Line 148-149]

b)        Z-scores for height-for-age were calculated using raw anthropometric measurements. Revised sentence for clarity [Page 4, Line 149-155]

Comments 3:

Data Analysis

a)      The logistic regression is appropriate for binary outcomes, but the selection process for included variables in the final model is not sufficiently described. Were variables chosen based on theoretical relevance or univariate significance?

b)      Consider reporting model fit statistics (e.g., Hosmer–Lemeshow test, AUC) to support the robustness of the regression models.

Response 3:

a)        Variables included in the final logistic regression model were primarily selected based on their univariate significance (p<0.05). Statement has been described in the manuscript. Revised sentence for clarity [Page 4, Line 166-167].

b)        Thank you for the valuable suggestion. In this study, we used complex sample analysis procedures. Thus, model fit statistics could not be performed. Therefore, the test was not applicable in our analysis.

Comments 4:

Interpretation of Results

a)        While several determinants are statistically significant, the authors should better distinguish between correlation and causation in the discussion.

b)        The interpretation of lower odds of stunting in older children needs clarification, especially in light of potential cumulative growth retardation over time.

c)          

Response 4:

a)        Discussion section – data interpretation has been revised accordingly [Page 13, Line 279-280; 298-300; Page 14, Line 323-324]

b)        Interpretation of results has been revised in Results Section 3.3 accordingly. [Page 5, Line 204-217] and clarified in the discussion [Page 13, Line 279-297]

Comments 5:

Policy Implications

The conclusion makes brief mention of the relevance to nutrition intervention programs. This could be expanded by referring to existing Malaysian policies or regional initiatives that could benefit from these findings

Response 5:

Thank you for the valuable suggestion. We have expanded the conclusion to highlight the relevance of our findings to existing nutrition intervention efforts in Malaysia [Page 15-16, Line 407-414; 420-429]

Comments 6:

Minor Comments

a)        In Table 1, please define abbreviations in footnotes (e.g., BMI-for-age z-score).

b)        A few typographical errors were found (“underweight” is occasionally misspelled).

c)         Consider improving the clarity of some sentences, especially in the results section where percentages and odds ratios are presented together—breaking them into separate statements can improve readability.

Response 6:

a)        Added accordingly in footnotes Table 1 [Page 8, Line 219-220]

b)        Revised accordingly.

c)         Thank you for your comment. We acknowledge the importance of clarity in presenting statistical findings. In this manuscript, we followed a standard reporting format for multivariable analysis, whereby the adjusted odds ratios (AORs), percentages, and 95% confidence intervals (CIs) are reported together to provide both descriptive context and inferential interpretation. This format is commonly used in epidemiological literature to facilitate comparison across variables. Therefore, no changes were made to the results section.

Additional clarifications:

We have made the following revisions to improve the manuscript:

a)      The language throughout the manuscript has been carefully revised for clarity and consistency, following American English conventions.

b)      The data in Table 3 have been reviewed and updated accordingly to ensure accuracy and completeness.

c)       Supplementary tables have been added to provide additional detail and improve clarity for the reader.

d)      List of Abbreviations has been updated to reflect all terms used in the manuscript.

e)       Added references have been updated and renumbered accordingly throughout the manuscript.

All changes have been highlighted and are visible using track changes in the resubmitted version of the manuscript.

Reviewer 2 Report

Comments and Suggestions for Authors

See Attached File

Author Response

1. Summary

2. Point-by-point response to Comments and Suggestions for Authors

Comments 1:

Overall Reviewer Comment:

This manuscript provides valuable insights into the risk factors associated with stunting, particularly focusing on demographic and household-level variables such as age, birthweight, household crowding, number of siblings, and race/ethnicity. While these are important predictors, the manuscript would be significantly strengthened by incorporating a broader range of modifiable socioeconomic and behavioral risk factors. Key omissions include dietary intake patterns (e.g., fish, milk, fruits), breastfeeding practices, parental education, household income, and maternal characteristics such as age, BMI, smoking, and alcohol use. Additionally, infection/inflammation markers, healthcare access, and other environmental or contextual factors (e.g., sanitation, hygiene) should be considered, given their established relevance to stunting. Inclusion of these modifiable and actionable variables would not only enhance the comprehensiveness of the analysis but also improve the utility of findings for informing public health interventions and policies. I encourage the authors to either incorporate these variables, if available, or clearly acknowledge their absence and discuss their potential influence in the limitations section.

Response 1: We sincerely thank the reviewer for the insightful feedback. In response, we have revised the objectives [Abstract section, Page 1, Line 17] and introduction [Page 2, Line 55-57; 70-83; 86; 88-93] to better reflect the broader range of modifiable and behavioral risk factors relevant to stunting, as suggested.

We have acknowledged in the revised limitations in discussion section [Page 15, Line 389- 396] that our dataset did not include variables such as dietary intake, breastfeeding practices, or infection/inflammation markers. We also discussed their potential influence on stunting risk and emphasized the need for future research to incorporate these factors for a more comprehensive understanding  

Comments 2:

Abstract

§   Bring Background/Objectives just after “Abstract”

Response 2: Thank you for your suggestion. Revised accordingly. [Page 1, Line 12-13]

Comments 3:

Introduction

The introduction is well-crafted, concise, and engaging.

a)       However, the authors should provide references for the statement: “Most studies focus on children under the age of five.” & “The complexity of the determinants of stunting is compounded by differences in access to healthcare, maternal employment, and economic conditions between urban and rural areas”.

b)       Further in second last paragraph of introduction, authors need to pull more studies to synthesize knowledge gap, regarding what risk factors already been evaluated to exhibit, what we already know and what we don’t as knowledge gap for broader range of modifiable socioeconomic and behavioral risk factors like dietary intake patterns (e.g., fish, milk, fruits), breastfeeding practices, parental education, household income, and maternal characteristics such as age, BMI, smoking, alcohol use. infection/inflammation markers, healthcare access, and other environmental or contextual factors (e.g., sanitation, hygiene), etc. “Addressing these gaps is critical to developing comprehensive life-cycle approaches that support children's growth from infancy through adolescence.”..and following paragraph is not well justified until comment 3 addressed.

Response 3:

We sincerely appreciate the reviewer’s kind remarks.

a)      We have added supporting citations for the claims about:

·         studies focusing on children under five [Page 2, Line 62-63]

12.   Bahtiar, Badriah Aisyah, Hayati Mohd Yusof, and Khairil Shazmin Kamarudin. "Child development and nutritional status of children under five: A cross-sectional study of a fishermen community in Terengganu, Malaysia." Jurnal Gizi Dan Pangan 16, no. 2 (2021): 91-100.

13.   Haron, Muhammad Zulfahmi, Abdul Jalil Rohana, Noor Aman A. Hamid, Mohd Azahadi Omar, and Noor Hashimah Abdullah. "Stunting and its associated factors among children below 5 years old on the east coast of peninsular Malaysia: evidence from the national health and morbidity survey." The Malaysian Journal of Medical Sciences: MJMS 30, no. 5 (2023): 155

14.   How, Eric Tan Chee, Suzana Shahar, Fredie Robinson, Abdul Marsudi bin Manah, Mohd Yusof Ibrahim, Mohammad Saffree Jeffree, and Syed Sharizman Syed Abdul Rahim. "Risk factors for undernutrition in children under five years of age in Tenom, Sabah, Malaysia." Malaysian Journal of Public Health Medicine 20, no. 1 (2020): 71-81.

15.   Logarajan, Renuka Devi, Norashidah Mohamed Nor, Saifuzzaman Ibrahim, and Rusmawati Said. "Social determinants of stunting in Malay children aged< 5 years in Malaysia." Nutrition 111 (2023): 112030.

16.   Rahayuwati, Laili, Maria Komariah, Citra Windani Mambang Sari, Desy Indra Yani, Yanti Hermayanti, Arlette Setiawan, Hediati Hastuti, Sidik Maulana, and Kelvin Kohar. "The influence of mother’s employment, family income, and expenditure on stunting among children under five: a cross-sectional study in Indonesia." Journal of multidisciplinary healthcare (2023): 2271-2278.

17.   Partap, Uttara, Elizabeth H. Young, Pascale Allotey, Manjinder S. Sandhu, and Daniel D. Reidpath. "Characterisation and correlates of stunting among Malaysian children and adolescents aged 6–19 years." Global health, epidemiology and genomics 4 (2019): e2.

·                     urban/rural disparities in stunting determinants [Page 2, Line 68-69]

15.   Logarajan, Renuka Devi, Norashidah Mohamed Nor, Saifuzzaman Ibrahim, and Rusmawati Said. "Social determinants of stunting in Malay children aged< 5 years in Malaysia." Nutrition 111 (2023): 112030.

19.   Siramaneerat, Issara, Erni Astutik, Farid Agushybana, Pimnapat Bhumkittipich, and Wanjai Lamprom. "Examining determinants of stunting in Urban and Rural Indonesian: a multilevel analysis using the population-based Indonesian family life survey (IFLS)." BMC public health 24, no. 1 (2024): 1371.

20.   Tadesse, Sisay Eshete, Tefera Chane Mekonnen, Reta Dewau, Aregash Abebayehu Zerga, Natnael Kebede, Yitbarek Wasihun Feleke, and Amare Muche. "Urban-rural disparity in stunting among Ethiopian children aged 6–59 months old: A multivariate decomposition analysis of 2019 Mini-EDHS." PloS one 18, no. 4 (2023): e0284382.

b)      We have expanded the introduction to better synthesize existing evidence on modifiable risk factors (diet, breastfeeding, parental education, etc.) and clarify remaining gaps [Page 2, Line 55-57; 70-83].

Added citation are as follows:

18.   Lee, Way Seah, Muhammad Yazid Jalaludin, Kim Mun Khoh, Juan Loong Kok, Thiyagar Nadarajaw, Anna Padmavathy Soosai, Firdaus Mukhtar et al. "Prevalence of undernutrition and associated factors in young children in Malaysia: A nationwide survey." Frontiers in pediatrics 10 (2022): 913850.

22.   Shariff, Zalilah Mohd, Khor Geok Lin, Sarina Sariman, Chin Yit Siew, Barakatun Nisak Mohd Yusof, Chan Yoke Mun, Huang Soo Lee, and Maznorila Mohamad. "Higher dietary energy density is associated with stunting but not overweight and obesity in a sample of urban Malaysian children." Ecology of Food and Nutrition 55, no. 4 (2016): 378-389.

23.   Cendana, Putri, and So-Young Kim. "Maternal Factors and Breastfeeding Practices Associated With Stunting Among Indonesian Children Aged 6 to 23 Months." Asia Pacific Journal of Public Health 37, no. 4 (2025): 402-410.

24.   Filiya, Ana Nur, Adenix Putri Ultasari, Novy Ardyanti Putri, and Aulia Afifah. "Correlation Between Exclusive Breastfeeding, Frequency and Quantity of Complementary Feeding With Stunting Among Toddler in Puru Village, Suruh District, Trenggalek Regency." Malaysian Journal of Medicine & Health Sciences 20 (2024).

25.   Setiani, Fibrinika Tuta, and Abdullah Azam Mustajab. "Faktor risiko stunting pada bayi dan balita (anak usia 0-59 bulan) di Wonosobo." Malahayati Nursing Journal 5, no. 7 (2023): 2134-2148.

26.   Mokalla, Thirupathi Reddy, and Vishnu Vardhana Rao Mendu. "Risk factors and socioeconomic inequalities in undernutrition among children 0-59 months of age in India." International Journal of Population Studies 5, no. 2 (2019): 14-23.

27.   Ahmed, Kedir Y., Abel F. Dadi, Felix Akpojene Ogbo, Andrew Page, Kingsley E. Agho, Temesgen Yihunie Akalu, Adhanom Gebreegziabher Baraki et al. "Population-modifiable risk factors associated with childhood stunting in sub-Saharan Africa." JAMA network open 6, no. 10 (2023): e2338321-e2338321.

28.   Muchlis, Nurmiati, Rezky Aulia Yusuf, Arni Rizqiani Rusydi, Nur Ulmy Mahmud, Nurul Hikmah, Andriany Qanitha, and Abdillah Ahsan. "Cigarette smoke exposure and stunting among under-five children in rural and poor families in Indonesia." Environmental health insights 17 (2023): 11786302231185210.

29.   Restila, Ridha, Bambang Wispriyono, Ririn Arminsih, Umar Fahmi Achmadi, Tri Yunis Miko, Defriman Djafri, and Miko Hananto. "Potential risk of stunting in children under five years living by the riverside: A systematic review." Malaysian Journal of Nutrition 29, no. 3 (2023).

30.   Prentice, Andrew M. "Environmental and Physiological Barriers to Child Growth and Development." In Nestle Nutrition Institute Workshop Series, vol. 93, pp. 125-132. 2020.

31.   Soliman, Ashraf T., Nada M. Alaaraj, and Alan D. Rogol. "The link between malnutrition, immunity, infection, inflammation and growth: New pathological mechanisms." World Journal of Advanced Research and Reviews 15, no. 1 (2022): 157-167.

Comments 4:

Methods:

1.       The manuscript mentions the use of a multistage cluster sampling procedure but does not provide sufficient detail on how this was implemented. To enhance transparency and reproducibility, the authors should briefly describe the stages involved in the sampling process (e.g., selection of clusters, households, individuals) and clarify how representativeness was ensured.

2.       ”The questionnaire on environmental factors consisted of 14 questions, that are known to affect children's health such as source of drinking water, sanitation, waste disposal methods, hand hygiene, and smoking in the home. Authors evaluated lots of factors, but included a few of them in analysis. Readers will be benefited, if authors can describe “the selection process of risk factors initially to include in questionnaires” & “to include in this study analysis” including citations.

3.       The authors use the standard WHO cutoff (HAZ <−2 SD) to define stunting in their study, as referenced from WHO AnthroPlus software. However, it is important to acknowledge the growing debate regarding the appropriateness of using a universal cutoff for defining stunting, particularly in diverse populations with varying genetic growth potentials and environmental contexts. Several studies have raised concerns that using a single global reference may lead to under- or overestimation of stunting prevalence in certain ethnic groups, potentially obscuring meaningful biological or social variation in child growth patterns (Grummer-Strawn et al., 2012; Leroy & Frongillo, 2019). Could the authors comment on how they interpret the applicability of the WHO growth standards in the context of their study population? In particular, do they believe population-specific references or adjustments would offer a more accurate reflection of growth deficits in this setting?

References for Authors' Consideration:

Grummer-Strawn LM, Reinold C, Krebs NF. Use of World Health Organization and CDC growth charts for children aged 0-59 months in the United States. MMWR Recomm Rep. 2010;59(RR-9):1–15.

Leroy JL, Frongillo EA. Perspective: What Does Stunting Really Mean? A Critical Review of the Evidence. Adv Nutr. 2019;10(2):196–204. doi:10.1093/advances/nmy101The cut off by WHO for stunting and other measures is being debated for ethnic group specific cut off, what is authors insights on this?

Response 4:

a)      We have revised the Methods section to provide a clearer description of the sampling stages. Kindly note that the full methodological details have been published elsewhere and have been referenced in the revised sentence to enhance clarity [Page 3, Lines 103 to 112].

b)      Thank you for your insightful comment. While we initially mentioned a total of 14 questions related to environmental factors, some of the questions were regroup to the main components for sanitation and hygiene practices in the analysis as clarified in the footnotes of Table 2. Regarding smoking habits (4 items), only one question related to whether parent smokes was included in the analysis. The remaining items were additional questions to provide more detailed understanding of smoking behavior but were not used in the final analytical model. Smoking in the home, however, was retained as an adjustment variable due to its potential confounding effect. We have added this information to clarify the results in Section 3.2 [Page 5, Lines 198 to 201].

a)        We acknowledge and appreciate the reviewer’s point, and it is indeed important to consider the applicability of growth standards in diverse populations. In this study, we used the WHO Child Growth Standards (2006) and the WHO Growth Reference (2007) to classify stunting among children aged 0.5 to 12 years. These references are commonly used in Malaysia for national surveys such as the National Health and Morbidity Survey (NHMS) [IPH, 2019] and nationally representative studies [Poh et al., 2013]. Currently, Malaysia does not have nationally established cut-offs or percentiles for HAZ. The use of WHO references supports standardisation in growth assessment and enables consistent comparisons across regional and national populations. Revised sentence as suggested for clarity. [Page 4, Line 148-152]

References:

Institute for Public Health. "National Health and Morbidity Survey 2019 Non-communicable disease, healthcare demand, and health literacy Key Findings." (2019).

Poh, Bee Koon, Boon Koon Ng, Mohd Din Siti Haslinda, Safii Nik Shanita, Jyh Eiin Wong, Siti Balkis Budin, Abd Talib Ruzita, Lai Oon Ng, Ilse Khouw, and A. Karim Norimah. "Nutritional status and dietary intakes of children aged 6 months to 12 years: findings of the Nutrition Survey of Malaysian Children (SEANUTS Malaysia)." British Journal of Nutrition 110, no. S3 (2013): S21-S35.

Comments 5:

Results:

a)       ..“Ethnicity was also found to be significantly associated with stunting, with Malay children exhibiting higher prevalence compared to children of Chinese and Indian ethnicities (p<0.01).”…, my comment #3 of methods may link this result?

b)       The manuscript would benefit from a deeper exploration of the potential role of chronic infection or inflammation as a pathway linking poor sanitation to stunting. If feasible, it would be valuable for the authors to evaluate infection or inflammation status using objective measures. For example, stool testing for helminth eggs or enteric pathogens (bacteria, protozoa) could provide direct evidence of gastrointestinal infections. In addition, including biomarkers of long-term infection or systemic inflammation—such as Erythrocyte Sedimentation Rate (ESR), C-reactive Protein (CRP), or other acute-phase reactants—could strengthen the biological plausibility of the hypothesized sanitation–infection–stunting pathway. If these data are available, the authors are encouraged to examine and report such associations. If not, a discussion of this limitation and the role of infection/inflammation as a plausible mediating mechanism would enhance the interpretation of findings.

c)       To improve the clarity and interpretability of the statistical analysis, I suggest that the authors first present the bivariate associations between each independent variable and stunting, indicating variables significantly associated (e.g., p < 0.05) in bold. Subsequently, those variables with significant bivariate associations only should be included in a single multivariable logistic regression model forcibly for mutual adjustment. It would be helpful if the adjusted odds ratios (AORs) from the multivariable model are presented in a table, with both significant and non-significant variables included for transparency. Please consider bolding statistically significant AORs (p < 0.05) for easy interpretation. This stepwise and comprehensive presentation would enhance the reader’s understanding of which variables independently predict stunting and how the associations change upon adjustment.

Response 5:

a)      Added point accordingly in the introduction part for ethnicity [Page 2, Line 55-57]

b)      We appreciate the reviewer’s valuable suggestion. As stool analysis and biomarker data were not available, we were unable to directly assess infection or inflammation in this study population. A paragraph has been added to the discussion to acknowledge this limitation [Page 15, Line 389-396]

c)       Thank you for the suggestion. We understand the importance of clearly presenting the relationship between each independent variable and stunting. While some of this information is already included in the regression table, we acknowledge that a separate table summarizing the univariate associations may improve readability and transparency. To address this, we have added two supplementary table (Supplementary Table 1 & 2) presenting the crude associations between each independent variable and stunting status. This allows readers to see the unadjusted relationships alongside the multivariable analysis presented in the main text.

Comments 6:

Discussion

a)       Provide a reference for the statement: “…In contrast, Chinese individuals are more likely to follow dietary habits aligned with health-promoting guidelines, such as higher consumption of vegetables, fruits, and lean proteins.…”if available...

b)       Add “which community” in Page 13, Line 284-286) “….

c)       Add more limitation as requested in earlier comments.

Response 6:

a)      A supporting reference has been added. [Page 14, Line 332]

b)      Added accordingly [Page 14, Line 344]

c)       Added accordingly [Page 15, Line 382-396]

Additional clarifications:

We have made the following revisions to improve the manuscript:

a)      The language throughout the manuscript has been carefully revised for clarity and consistency, following American English conventions.

b)      The data in Table 3 have been reviewed and updated accordingly to ensure accuracy and completeness.

c)       Supplementary tables have been added to provide additional detail and improve clarity for the reader.

d)      List of Abbreviations has been updated to reflect all terms used in the manuscript.

e)       Added references have been updated and renumbered accordingly throughout the manuscript.

f)        All changes have been highlighted and are visible using track changes in the resubmitted version of the manuscript.

Round 2

Reviewer 2 Report

Comments and Suggestions for Authors

MS: Nutrients-3700671_R1

Title: Determinants of Stunting among Children aged 0.5 To 12 Years 2 in Peninsular Malaysia: Findings from SEANUTS II Study

Comments for authors

I commend the authors on their effort to respond to reviewers' comments and, overall, they adequately addressed my concerns.  

Hence, I recommend for consideration of publication of this manuscript.

Author Response

Comments for authors:

I commend the authors on their effort to respond to reviewers' comments and, overall, they adequately addressed my concerns.  

Hence, I recommend for consideration of publication of this manuscript.

Responses:

We sincerely thank the reviewer for the thoughtful and constructive feedback provided throughout the review process. We truly appreciate your recommendation for the manuscript to be considered for publication.